# PLA Reinforced with Limestone Waste: A Way to Sustainable Polymer Composites

**DOI:** 10.3390/polym17050662

**Published:** 2025-02-28

**Authors:** Dora Sousa, Catarina Baleia, Pedro Amaral

**Affiliations:** Instituto de Engenharia Mecânica (IDMEC), Instituto Superior Técnico, Av. Rovisco Pais 1, 1049-001 Lisboa, Portugal; catarina.baleia@tecnico.ulisboa.pt (C.B.); pedro.amaral@tecnico.ulisboa.pt (P.A.)

**Keywords:** PLA extrusion, 3D printing, limestone, stone waste, recycling, biopolymers, filament

## Abstract

Waste stone sludge generated by the extractive industry has traditionally posed significant disposal challenges. This study redefines stone sludge as a valuable raw material by incorporating it into polylactic acid (PLA) to create sustainable composite materials. Pellets and filaments composed of up to 50% by weight of limestone powder and PLA were successfully produced using melt blending in a twin-screw extruder. Scanning electron microscopy (SEM), X-ray fluorescence (XRF), and X-ray diffraction (XRD) analyses revealed a uniform distribution of stone particles within the PLA matrix and confirmed the chemical and structural compatibility of the components. Thermogravimetric analysis (TGA) showed that the composites retained thermal stability, while mechanical testing demonstrated significant enhancements in stiffness, with an increase in elastic modulus for composites containing 50% limestone powder. The melt flow rate (MFR) decreases with increasing filler content. The brittleness also increased, reducing impact resistance. Mechanical tests were performed on injected and 3D-printed specimens. The filament produced was successfully used in 3D printing, with a small XYZ calibration cube.

## 1. Introduction

This study explores the use of limestone waste as a filler in PLA to develop sustainable polymer composites for various applications, including 3D printing and injection molding. The novelty of this research lies in maximizing the substitution of the polymer content with limestone powder—up to 50% by weight—while maintaining the composite’s essential properties and functionality as a polymer. To further enhance the material’s performance and compatibility, additional additives such as plasticizers and compatibilizers can be incorporated. This approach not only improves sustainability by reducing reliance on virgin polymers but also ensures the material’s versatility for use in diverse manufacturing processes beyond FDM, such as traditional injection molding.

Recently, environmental awareness has risen, leading to an increasing demand for more sustainable and alternative solutions in the production of composite materials. A promising innovation in this field is the use of mineral-reinforced polymers, which utilize natural minerals as reinforcement agents. Natural minerals are crucial for enhancing the strength and toughness of mineral-reinforced polymers [1].

The goal is to leverage endogenous resources, such as the limestone deposits found in the central region of Portugal, for cutting-edge technologies like additive manufacturing. Our research focuses on developing materials and innovative compounds through the transformation of extracted natural limestone. These materials are to be applied in advanced additive manufacturing technologies, specifically through extrusion-based processes with thermoplastic matrix compounds like PLA.

With proper treatment and characterization, these “secondary materials” can find renewed purpose in innovative 3D printing processes. This opens opportunities for the creation of shapes and designs that were previously unattainable.

Solid stone wastes come from the rejections at the quarry sites or at the processing facilities, which can be in the form of fragments/slabs and stone sludge. Stone sludge is a semi-liquid fluid made up of stone grains (usually smaller than 100 microns and rarely larger than 150) mixed with water.

This waste is mainly used in the construction sector as an aggregate in concrete and asphalt, as it enhances the durability and stability of these construction materials, reducing costs and conserving natural resources, and as a filler in concrete mixtures, as it improves workability, reduces the need for higher-cost binders, and contributes to more sustainable construction practices [2,3,4], and also in agriculture as a soil stabilizer [5,6].

Use of Stone Waste as a Mineral Filler:

Limestone is a sedimentary rock mostly composed of calcium carbonate (CaCO_3_) in the form of the mineral calcite. It is one of the most abundant and versatile types of rock found on the Earth’s surface. Limestone forms through the accumulation of marine organisms, such as coral, shells, and microorganisms, and the precipitation of calcium carbonate from water [7].

Limestone has essentially the same chemical and mineralogical composition as marble, apart from the ones with a great number of minerals formed by impurities. Thus, it is also composed primarily of calcite [8].

Esposito Corcione et al. have recently used 3D printing technology in conjunction with stone waste to craft an innovative composite biomaterial. This material is a mixture of PLA and discarded Lecce Stone (LS) scraps. Their study aimed to ascertain its applicability in the hands of designers for the creation of original industrial and construction products, offering a means to fashion a wide array of shapes and geometries efficiently and without wastage. In essence, this paper serves as a proof of concept, attesting to the suitability of LS waste as a potent filler for high-strength bio-composite polymers [9].

Pavon et al. studied the effect of CaCO_3_ fillers extracted from waste eggshells on 3D-printed PLA performance [10] and examined calcium carbonate fillers derived from natural resources but focused on lower filler concentrations and their general impact on PLA.

Lendvai et al. researched the potential of incorporating marble dust as a filler within PLA [11].

Awad et al. explored the potential of employing Egyptian marble dust particles as reinforcements in polymeric composites, particularly polypropylene (PP). This significantly improved both thermal and mechanical properties, including the flexural strength, compressive strength, elastic modulus, and hardness of the composites [12].

Kashyap and Datta [13] used limestone sludge-filled composites of HDPE-PP blends to study the effects of limestone sludge addition on the mechanical, thermal, and morphological properties of HDPE-PP blends. They concluded that limestone sludge in its calcite phase is thermally very stable to be used in a polymeric matrix.

Sawalha and colleagues conducted research involving the compounding of polyethylene with stone waste through an extrusion process, and they established a direct relationship between the enhancement of polymer properties and the key aspects of filler particles [14].

Doan et al. investigated the use of ground fragments and slurry waste materials (50–65%) as fillers for unsaturated polyester resin (UPE). This not only leads to cost-effective materials but also contributes to waste reduction [15].

Other applications have been studied, such as food packaging [16] and foams [17].

Fajdek-Bieda and Wróblewska reviewed potential applications of mineral-reinforced polymers with minerals in various industrial sectors, including packaging, automotive, construction, and medicine [1].

The transition to a circular materials economy requires biobased, biodegradable plastics that are recyclable and degrade in the environment [18].

Similar reinforced polymeric products exist on the market; for example, the KLONER3D^®^ company (IT) sells a filament for FFF in gypsum and harmless co-polyesters called LAYBRICK, which has the appearance of a stone; the Emerging Objects company (USA) prints 3D objects in sand, stone, and cement; and the FormFutura company (NE) commercializes filaments (STONEFIL) based on PLA and stone (50%), which contains the options of cement, clay, terracotta, and granite [19].

Additive Manufacturing:

Fused Deposition Modeling, or FDM, emerged in the 1980s, and in the 1990s, it was commercialized by Stratasys Inc., USA [20]. Its widespread adoption is attributed to its proven reliability, cost-effectiveness in manufacturing 3D objects with high resolution, exceptional dimensional stability, broad material customization options, straightforward fabrication procedures, and capacity to safely produce intricate geometric components in favorable working conditions. These attributes collectively contribute to FDM’s continued status as one of the most prevalent AM processes [21].

According to the American Society for Testing Materials (ASTM) international standard, AM is a process of joining materials to make objects from 3D model data layer by layer, as opposed to subtractive manufacturing methodologies, such as traditional machining [22].

Crucially, the characteristics of the end product, including its strength, surface finish, and porosity, are dependent on the specific process parameters employed [21].

Polylactic Acid:

Given that PLA is one of the prominent biodegradable synthetic polymers, there is significant interest in researching the integration of diverse fillers to enhance both the mechanical characteristics and the biodegradability of commercially accessible polymers [23]. It has been used widely because of its sustainability, non-toxicity, and biocompatibility in several applications, such as papermaking, biofuels, and biocomposite materials [19]. Also, as reviewed by Giammaria, it can be applied in the automotive sector [24].

PLA is a synthetic polymer made from natural, renewable materials like corn. While PLA is often promoted as a sustainable alternative to traditional plastics due to its biobased origins, its environmental benefits are limited by the specific conditions required for proper composting. PLA requires industrial composting facilities with high temperatures and controlled environments to break down efficiently [25]. In typical landfill conditions, PLA degrades very slowly, with studies indicating that only about 1% decomposes over 100 years [26]. However, unlike conventional plastics, PLA is not derived from fossil fuels, making it a more sustainable choice in terms of raw material sourcing. Additionally, its durability, evidenced by its resistance to degradation in landfill conditions, offers advantages for applications requiring long-term stability.

One of the advantages of PLA is its low melting point, which is one of the supportive parameters in 3D printing, as it consumes less energy when compared with the other commonly used polymers, like ABS and polyamides. Moreover, PLA is considered a user-friendly material that can be processed without producing toxic fumes [27].

The use of waste conventional plastic to develop 3D parts is an emerging area in which the waste plastics can be mixed with biopolymers and/or biomass (as reinforcement) to produce high-performance sustainable polymer composites with enhanced properties to suit the targeted applications. In this regard, an important key factor to create new and more sustainable materials is the use of recycled plastics and bioplastics to produce composites to reduce the carbon footprint of materials and the dependence on non-renewable raw materials [28].

## 2. Materials and Methods

The experimental work was designed to study the development of a compound of PLA-limestone powder (PLA-LP) and its applicability on additive manufacturing applications. Mixtures of up to 50 wt. % of stone powder were prepared and thoroughly studied. The main experimental methods are described below.

### 2.1. Materials

The PLA used as polymer matrix was the INZEA F2 HTS 451 purchased from Inzea Biopolymers (Zaragoza, Spain). It has a melting temperature of 175 °C, an MVR of 40 g/10min (measured at 210 °C with a 2.16 kg load), and a density of 1.32 g/cm^3^.

The limestone powder (LP) was supplied by the Filstone, Comércio de Rochas, SA (Fátima, Portugal). The stone waste retrieved appeared damp, and so it was dried in an oven at 60 °C for two days to reduce the water content. The dried powder agglomerates were crushed with a mortar and pestle and then sieved in a sieve with a fine mesh with a diameter of 355 µm.

### 2.2. Characterization

SEM was used to characterize the limestone powder and the cross-section of the filaments. This technique was used coupled with an energy-dispersive spectrometer (EDS). Samples were placed on an Al stub using double-sided carbon tape and sputter coated with a thin Au/Pd film in a Quorum Technologies coater, model Q150T ES. They were then analyzed in a Hitachi scanning electron microscope (SEM), model S2400, equipped with a SDD light elements energy dispersive spectroscopy (EDS) Bruker detector. The acceleration voltage was 10 kV, and samples were coated with gold/palladium.

XRF is used for elemental and chemical analysis by measuring the emission of characteristic secondary or X-rays from material that has been excited by high energy X-rays. This analysis can be used for rapid analysis of the sample for regular quality control. The powder samples were measured on a Bruker S4 Pioneer wavelength dispersive X-ray fluorescence spectrometer, (Karlsruhe, Germany). Seven grams of each powdered sample was used for the analysis.

XRD analysis was used to evaluate the structure of crystalline substances and determine the mineralogical composition of the powdered samples. The X-ray diffractograms were performed using a XPERT-PRO (PANalytical, Almelo, The Netherlands) diffractometer with Cu Kα radiation and a X’Celerator detector. The measurement conditions were as follows: 40 kV and 35 mA, 0.033° 2θ step size and scan step time of 74.0167 s. The PANalytical “High Score Plus” software and PDF2 database were used.

TGA analysis was performed to quantify the amount of stone powder and polymeric binder on the composite material. TGA HITACHI STA 7200 equipment was used to conduct the analyses that were carried out in a controlled nitrogen atmosphere at a temperature increase rate of 10 °C·min^−1^, from room temperature to 600 °C.

MFR was determined on a UnitedTest MFI 405 Melt Flow Indexer in accordance with ASTM D1238 using a piston load of 2.16 kg at a barrel temperature of 190 °C.

### 2.3. Preparation of Pellets of PLA-Limestone Powder (PLA-LP)

The melt mixing and extrusion of the PLA-based composite containing up to 50 wt. % of LP was achieved using a twin-screw extruder machine (Werner and Pfleiderer-ZSK25x38D, Periplast).

The equipment consists of two screws, a motor with a power of 10 kW and a maximum screw speed of 300 rpm. The machine consists of a tank equipped with rollers and water for cooling the filament material and a granulator that allows the material to be cut into pellets.

PLA-LP mixture was extruded using a temperature profile in Table 1.

The extruded filaments were successively cut in the pelletizer. After dehydration in an oven at 50 °C for 12 h, the PLA-LP pellets were stored in sealed plastic bags and conditioned at room temperature for further investigation.

Although alternative polymers are plausible choices, PLA was favored for its manageable characteristics, attributed to its low melting point and degree of crystallinity. Moreover, recycled PLA was also incorporated into the process to produce a second sort of pellets.

### 2.4. Preparation of PLA-Limestone Powder Filament

Both pure PLA and PLA-LP pellets (with up to 50% by weight of stone powder) were processed through melt extrusion to produce filaments suitable for Fused Deposition Modeling (FDM). The extrusion was performed using a 3devo Composer 450 filament maker, a versatile machine with four heating zones capable of reaching temperatures up to 450 °C. The machine includes a nitride-hardened steel single-screw extruder, a replaceable nozzle, a dual air-cooling system with adjustable fan speeds and positions, an optical sensor, and a dynamic puller system, ensuring a precise filament diameter tolerance of ±0.05 mm, depending on the material used (Figure 1)**.**

After extrusion, the filaments were dried at 50 °C for an additional six hours. To achieve a consistent filament diameter of 1.75 mm, required for the 3D printing equipment, distinct temperature profiles and fan speeds were used for pure PLA and PLA/LP composites (Table 2). The machine’s preset temperature settings for PLA were adopted for this process.

### 2.5. Injection of Test Pieces

Specimens were prepared by injection molding using an injection molding machine (Boy 22A), a melt temperature of 210 °C, an injection rate of 40 cm^3^/s, and a residual cooling time of 15 s for PLA-LP composites and 35 s for PLA.

### 2.6. Printed Test Pieces

For the printed specimens, a Creality Ender 3 V2 printer was used, the gcode was generated by PrusaSlicer 2.9.0. with the same dimensions as the injected pieces (see below).

The parameters used were nozzle diameter, 0.8 mm; infill density, 100%; raster angle, 45°, bed temperature, 60 °C. An extrusion temperature of 238 °C was used for all percentages.

### 2.7. Mechanical Testing

Using a 59R5884 with a video extensometer Instron AVE (Instron Ltd., Norwood, MA, USA) fitted with a 10 kN force sensor, the tensile characteristics of the samples were investigated. Tensile tests were performed on dumbbell-shaped specimens that had a cross-section of 10 mm by 4 mm (according to EN ISO 527-2 [29]). A crosshead speed of 10 mm/min and a clamping distance of 110 mm were established. The results presented are the averages of five separate, concurrent measurements made at room temperature.

Charpy impact tests were performed on an Instron Ceast 9050 pendulum-type impact tester, according to the EN ISO 179 standard [30,31]. The bearing distance was set to 62 mm, and the impact energy was 1 J. The results reported are the averages of five separate, concurrent measurements made at room temperature.

## 3. Results

### 3.1. Raw Material Characterization

The stone sludge generated after the diamond wire sawing step from the Filstone quarry was collected and fully characterized.

Multiple SEM images of the limestone powder allowed the identification of the grain’s morphology and dimension. Images were captured at different magnifications to offer a comprehensive view of the particles across various size ranges. A total of 300 manual measurements of the major axis length of the particles were conducted across different areas of the sample, ensuring a thorough representation of the stone powder’s particle size distribution. Figure 2 demonstrates that the particle diameter after sieving is within the range of 0–10 µm.

This technique provides semi-quantitative information on the amounts of certain elements in the sample. These quantities can be expressed in elemental or oxide form (normally, in the field of geology, they are expressed in oxide form, as in this case), but it does not imply that they are in this form. The values listed in the column “Others” in Table 3, corresponds to the additional elements percentage that could not be detected by the equipment.

From the XRD we concluded that the calcium oxide found in XRF is a form of calcite. This is common for the limestone found in this area of Portugal.

The resultant PLA-LP pellets of 50/50 % wt. consist of virgin PLA with limestone powder (Figure 3a) and recycled PLA parts with limestone powder (Figure 3b), respectively.

The TGA and DTG curves reported in Figure 4 confirmed the presence of about 50 wt. % of mineral content in the composite; for PLA, this value is zero since there is no addition of limestone powder. The discrepancy observed in the TGA and the results is attributed to the presence of unidentified additives in the Inzea PLA, designed to reduce its abrasiveness. These additives influence the TGA curve. However, this variation was carefully accounted for during the calculation of the stone content within the polymeric matrix, ensuring the results remain accurate and representative. Furthermore, the TGA curves show the degradation temperature for PLA is 347 °C and for 50 wt. % PLA/LP is 310 °C; thus, there is a slight reduction in the degradation temperature with the addition of 50% limestone powder.

Reference [9] claims the TGA curves of both the sample tested evidenced that the presence of LP does not influence the PLA degradation temperature, with the latter remaining almost unchanged and equal to 170 °C. This result indicates that the processability of the PLA filament using the FDM technique will not be compromised by the presence of the inorganic filler.

As expected, in Table 4, the MFR and melt volume rate (MVR) of PLA decrease as the concentration of CaCO_3_ increases due to higher melt viscosity. The degree of reduction depends on the filler’s particle size, shape, and concentration, as well as any interface modifications.

### 3.2. PLA-Limestone Powder Filament for FDM

With the aim of creating a stone filament suitable for 3D printing, a blend of PLA and stone powder (up to 50% by weight) was utilized. The filament underwent manufacturing via melt blending facilitated by a 3devo filament maker.

SEM was employed to examine the fracture surfaces of PLA-limestone composite filaments with a limestone content of 50 wt. %. This analysis aimed to assess the presence of defects, including air inclusions, scrutinize sample morphology, and analyze the distribution of stone particles within the PLA matrix (Figure 5c). The image reveals excellent dispersion of both polymers and powder. Additionally, the grain size of the powder remains within the filament’s diameter.

The extruded filament, virgin PLA-LP 50 wt. %, and recycled PLA-LP 50 wt. % are shown in Figure 5a and Figure 5b, respectively.

The filament consisting of a 50 wt. % weight ratio was effectively employed on a 3D printer (Creality Ender 3 V2). Due to its brittle nature, the filament is maintained at approximately 20 °C within a chamber just prior to its entry into the curved tube. A cube printed using this material is shown in Figure 5d.

The stone filament was processed at temperatures ranging from 220 to 230 °C, with a bed temperature set between 50 and 60 °C, a print speed of 30 mm/s, and 100% infill.

### 3.3. Mechanical Testing

The tensile mechanical properties of pure PLA specimens and its composites with 25 wt. % and 50 wt. % of LP are reported in Table 5. It is possible to perceive that the stone particles modified the mechanical behavior of the PLA matrix.

The term “tensile strain at tensile strength” represents the strain value observed when the material reaches its ultimate tensile strength (UTS), which is the maximum stress the material can withstand during the test. For ductile materials, this point typically corresponds to the highest peak on the stress–strain curve. However, for brittle materials, such as PLA-LP composites, the stress–strain curve does not exhibit a pronounced peak or yielding phase. Instead, it shows a linear elastic response until failure. In this case, the tensile strain at tensile strength corresponds to the strain value recorded at the point of maximum stress before the material fractures.

The tensile strength of the PLA composites (injected) decreased by 15% and 25% with the addition of 25 wt. % and 50 wt. % LP, respectively. This reduction can be attributed to two key factors: particle agglomeration and insufficient interfacial adhesion between the stone particles and the polymer matrix [32]. However, the addition of LP significantly enhanced the material’s stiffness. The incorporation of 25 wt. % LP resulted in an 8% increase in Young’s Modulus, while 50 wt. % LP led to a 61% increase. This improvement is due to the highly rigid stone particles acting as stiff reinforcements, which restrict the mobility of the polymer chains, thereby increasing resistance to deformation under stress [33].

The tensile strain at break decreased from 7% for pure PLA to 2.06% and 0.98% for composites with 25 wt. % and 50 wt. % LP, respectively. This indicates increased brittleness in the material. The stress–strain behavior (Figure 6a) shows a linear elastic response up to failure, without yielding or plastic deformation. The high filler content hinders the mobility of polymer chains, making the material less capable of plastic deformation. Consequently, the material absorbs applied energy without significant deformation and fractures readily.

Despite the reduction in tensile strength to 34 MPa at 50 wt. % LP, the material maintains considerable stiffness and strength, making it suitable for structural applications where minimal permanent deformation is required. These properties highlight the potential of PLA-LP composites as robust and durable materials for use in specific engineering and structural contexts.

The results indicate that the mechanical properties of the 3D-printed specimens are generally lower than those of the injected ones, particularly in terms of elastic modulus and tensile strength.

The lower mechanical properties observed in printed specimens compared to injected ones can be attributed to weaker interlayer adhesion, increased porosity, reduced polymer chain orientation, and differences in crystallization behavior.

In injection molding, parts are produced with higher density and stronger intermolecular bonding due to the combination of high pressure and controlled cooling conditions. This results in superior mechanical properties and fewer defects compared to 3D printing. In contrast, the layer-by-layer deposition process of 3D printing relies on localized heat and material diffusion for bonding between layers, which is generally weaker than the molecular cohesion achieved in injection molding. This reduced adhesion can lead to diminished mechanical performance, particularly in long printed parts like those in this study, where ensuring strong bonding between successive layers becomes challenging.

3D printing also introduces anisotropy, where the mechanical properties vary depending on the printing orientation. The bonding between layers is typically weaker than the strength within a single layer, leading to lower overall mechanical resistance when compared with injection where the material is processed as a continuous mass, ensuring uniform mechanical properties throughout the part.

The data presented in Table 6 shows for the injected specimens that the pure PLA matrix exhibited an energy absorption of 10.098%, with a residual strength (Re) of 2.75 kJ/m^2^. The addition of limestone to PLA led to a substantial reduction in energy absorption capacity and impact resistance. For example, with 25% wt. %, the energy absorption dropped to 8.26%, and the Re decreased to 2.06 kJ/m^2^. This trend continued with 50 wt. %, as energy absorption was further reduced to 5.998%, and Re was reduced to 1.514 kJ/m^2^.

The elastic modulus (E) values in Table 5 and the Absorbed Energy (%) values in Table 6 are consistent with the expected behavior of PLA-LP composites. The addition of limestone powder increases stiffness (higher E) due to the rigid reinforcement provided by the filler. However, this also reduces the material’s ability to absorb energy (lower Abs. En. %), as the composite becomes more brittle. This trade-off between stiffness and toughness is a characteristic of composites with high filler content and aligns with the observed data.

## 4. Discussion

A composite of limestone powder and PLA can be used to produce a variety of products such as bricks, wall panels, flooring tiles, outdoor furniture, decorative elements, sunglasses, among others, and basically any product that is produced today from 100% plastic can be produced with these composites. These can be lightweight and have insulating properties. They can be personalized, and as many or as few as needed can be produced.

The specific application and properties of the composite will depend on the ratio of limestone to PLA, the manufacturing process, and any additives or reinforcements used.

The combination of PLA and limestone produces a composite that is particularly lighter than traditional natural stones, yet equally robust. This offers significant benefits for construction, reducing the weight of structures without compromising durability. Additionally, the composite is malleable, allowing for a variety of customized applications, from cladding to decorative elements.

Another crucial aspect is the versatility of the PLA and limestone composites, as they can be used in additive manufacturing as a filament and also in pellet form. Composites of limestone powder and PLA in the form of pellets are easily integrated in the industry since they can be processed in well-established conventional polymer technologies.

The plastics industry can benefit by creating more sustainable, lightweight products when compared to metal or stone products, reducing waste and carbon emissions associated with transportation.

In summary, the 50/50 wt.% PLA and limestone composite signifies a substantial advancement for both the plastics and stone industries. This composite merges the versatility of PLA with the abundant availability of limestone, offering a more environmentally friendly and sustainable alternative. Adopting this innovation can reduce the industry’s carbon footprint while enabling the development of eco-friendly, durable products, thereby contributing to a greener and more sustainable future.

At present, limestone powder, which is extracted from the residue of stone sawing and processing slurry, has yet to discover economically viable industrial applications. Research efforts have primarily concentrated on repurposing micro-fine sawdust from marble in cost-effective applications. A more sustainable approach involves harnessing the microfine dust as a valuable by-product. This approach allows companies to offset landfill expenses by generating revenue through the sale of these materials [34].

## Figures and Tables

**Figure 1 polymers-17-00662-f001:**
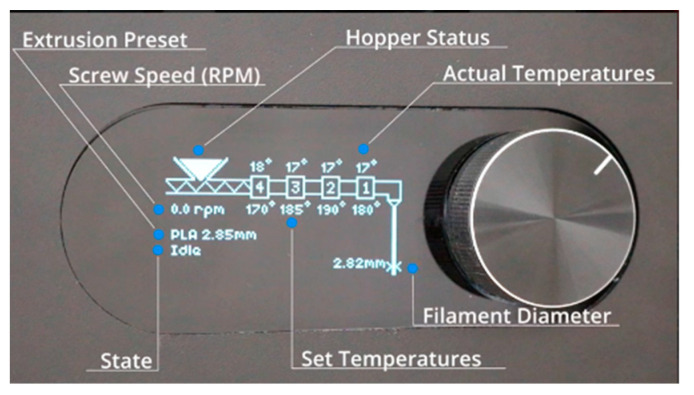
3devo filament maker schematics from the manual.

**Figure 2 polymers-17-00662-f002:**
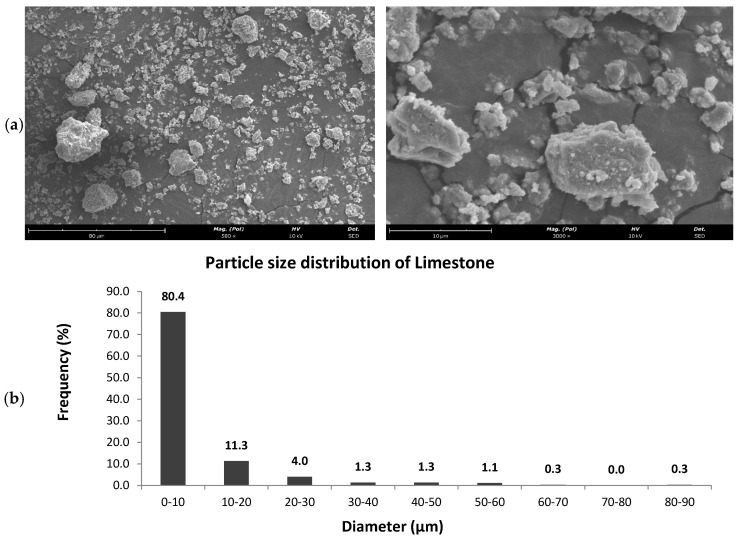
SEM images of the utilized limestone powder (**a**). Size distribution in micrometers for the limestone powder (**b**).

**Figure 3 polymers-17-00662-f003:**
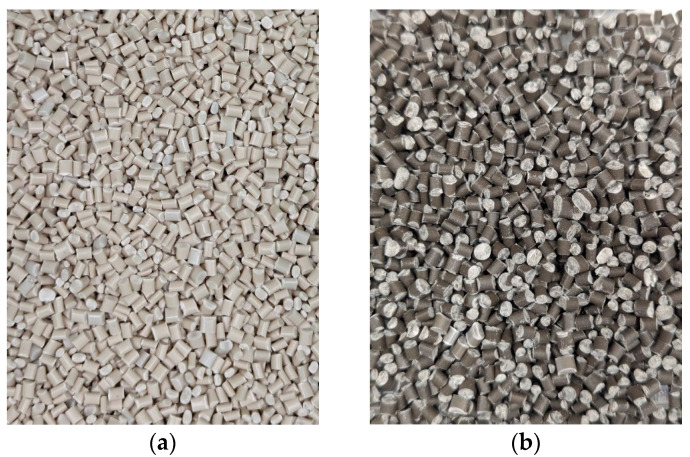
Pellets of the compound of PLA and limestone powder, (**a**) from virgin PLA, (**b**) from PLA recycled parts.

**Figure 4 polymers-17-00662-f004:**
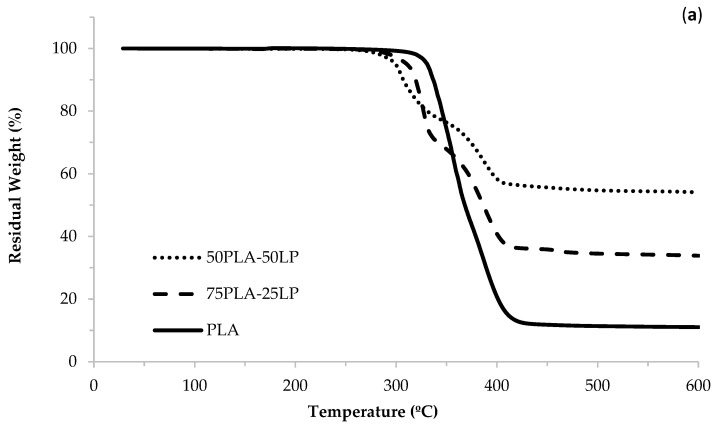
(**a**) TGA thermograms, (**b**) DTG thermograms for PLA and composites.

**Figure 5 polymers-17-00662-f005:**
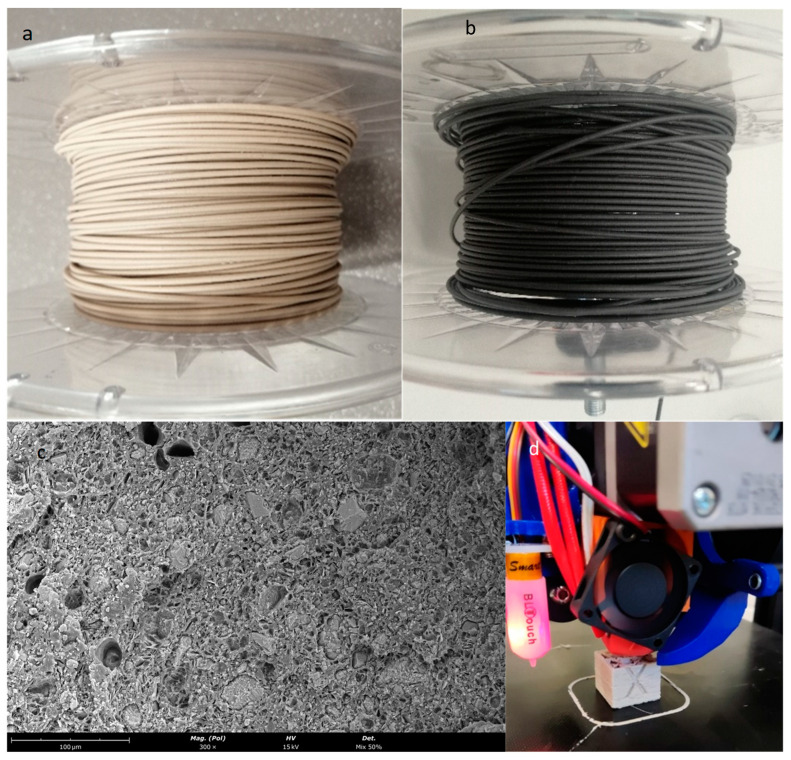
(**a**) PLA filament of virgin PLA-LP, (**b**) recycled PLA-LP, (**c**) SEM image of the cross-section of the PLA-LP compound and (**d**) cube printed with a filament of 50 wt. % PLA/LP.

**Figure 6 polymers-17-00662-f006:**
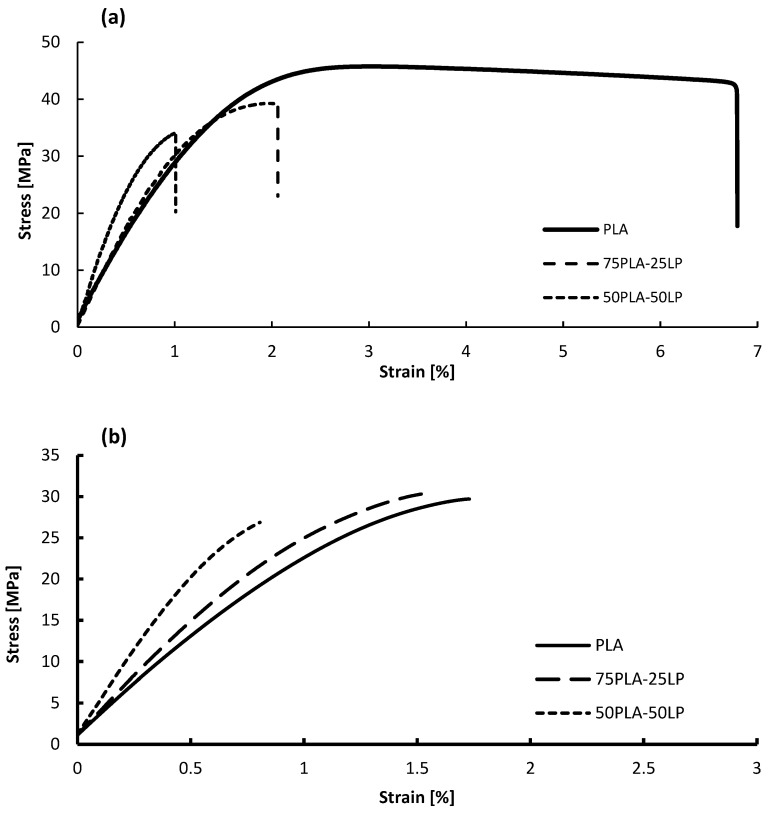
Stress–strain curve for PLA and composites, injected (**a**) and printed (**b**). Attention to scale for comparison.

**Table 1 polymers-17-00662-t001:** Parameters for the Werner and Pfleiderer extruder.

Heater 1	Heater 2	Heater 3	Heater 4	Heater 5	Heater 6	Heater 7	Heater 8	Heater 9	Rotation (rpm)
180 °C	177 °C	175 °C	170 °C	168 °C	173 °C	165 °C	168 °C	168 °C	60–70

**Table 2 polymers-17-00662-t002:** Settings for the 3devo filament maker.

Settings	PLA	PLA/LP
Heater 1	170 °C	170 °C
Heater 2	185 °C	180 °C
Heater 3	190 °C	190 °C
Heater 4	180 °C	180 °C
Extrusion speed	3.5 rpm	3.5 rpm
Filament fan speed	70%	20%

**Table 3 polymers-17-00662-t003:** XRF results for the limestone sludge sample from Filstone.

Sample	CaO	MgO	P_2_O_5_	Others
Filstone	72.3	0.7	0.4	22.6

**Table 4 polymers-17-00662-t004:** MVR and MFR values for PLA and composites.

Sample	Capture Time (s)	MVR (cm^3^/10 min)	MFR (g/10 min)
PLA	8.34	32.52	42.92
75PLA-25LP	13.70	19.79	26.12
50PLA-50LP	33.01	8.22	10.84

**Table 5 polymers-17-00662-t005:** Tensile test results for PLA/LP.

Method	Samples	Elastic Modulus (E)	Tensile Strength	Tensile Strain at Tensile Strength	Tensile Stress at Break	Tensile Strain at Break
		(MPa)	(MPa)	(%)	(MPa)	(%)
Injected	PLA	3143 ± 45	45.6 ± 0.3	3.38 ± 0.48	42.5 ± 0.6	7 ± 1.2
	75PLA-25LP	3403 ± 100	38.5 ± 0.5	2.00 ± 0.00	38.2 ± 0.5	2.06 ± 0.05
	50PLA-50LP	5069 ± 193	34.0 ± 0.8	0.98 ± 0.08	34.0 ± 0.7	0.98 ± 0.08
Printed	PLA	2478 ± 70	30.1 ± 1.5	1.70 ± 0.1	30.0 ± 1.6	1.70 ± 0.1
	75PLA-25LP	3010 ± 130	30.3 ± 1.9	1.50 ± 0.09	30.3 ± 2.0	1.60 ± 0.1
	50PLA-50LP	4208 ± 169	26.8 ± 1.8	0.80 ± 0.08	26.8 ± 1.8	0.80 ± 0.1

**Table 6 polymers-17-00662-t006:** Charpy results for PLA and composites.

	L (mm)	W (mm)	T (mm)	Abs. En. (%)	Re (KJ/m^2^)
PLA	81.106	9.7524	3.753	10.09	2.75
75PLA-25LP	81.426	9.8294	3.918	8.26	2.06
50PLA-50LP	79.424	9.864	3.9414	5.99	1.51

## Data Availability

The original contributions presented in this study are included in the article. Further inquiries can be directed to the corresponding author.

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
