# Peer review of "PLA Reinforced with Limestone Waste: A Way to Sustainable Polymer Composites"

_polymers, 2025, doi:10.3390/polym17050662_

Round 1

Reviewer 1 Report

Comments and Suggestions for Authors

Novelty:

The article does not include a statement regarding its novelty. The authors should explicitly delineate how they intend to assert the novelty of their work, particularly in relation to references 13 and 14, and, most importantly, in comparison to reference number 15. 

Abstract: 

Authors should enhance the content of the abstract by incorporating results to emphasise the significance of the study.

Introduction:

1. The introduction seems to be lacking in completeness, as it appears that the authors have delineated various segments without adequately emphasising the novelty of the study.  

2. The inclusion of subsection headings appears unnecessary; therefore, the authors are advised to eliminate the sub-headings 1.1, 1.2, and 1.3.

3. Several assertions have been made without the backing of a literature review.

4. The number of cluster references (2-6 & 7-10) should be decreased.

5. Abbreviations should be defined upon first use, after which they can be utilised instead of the full form. For example PLA, LP, Scanning electron microscopy, and much more

Characterization:

1. Procedural details should be presented clearly, ensuring that readers can comprehend them effectively, SEM and X-ray fluorescence spectrometer 

Results:

1. The TGA of LP is absent. Authors are required to include this data in the manuscript for comparative assessments of composite materials.

2. Page 7, line 276-279: TGA explanation is not matching with results in Figure 3.  

3. What is MVR?

4. What does authors mean "Tensile strain at tensile strenght

5. "Elastic Modulus (E)" in Table 4 are not in accordance with "Abs. En. (%)" in Table 5.

6. Authors needs to rephrase a large part for easy comprehension of results

Comments on the Quality of English Language

Must be improved

Reviewer 2 Report

Comments and Suggestions for Authors

Major revisions are required before acceptance for publication in Polymers. Below are my comments:

1.In the introduction, please describe a history of other natural fibres in the additive manufacturing process. For instance, similar studies such as the following research[*], showed that basalt fibre produced from basalt rocks, can be employed as a reinforcement in recycling and upcycling process of thermoplastic materials.

*Ghabezi, P., et al. (2024). Upcycling waste polypropylene with basalt fibre reinforcement enhancing additive manufacturing feedstock for advanced mechanical performance. Applied Materials Today41, 102486.

2. In Section 1.3, Polylactic Acid (PLA) is made from natural, renewable materials like corn, but that doesn’t mean it’s sustainable. The main problem is that PLA needs very specific conditions to compost properly. In landfills, it stays mostly intact, with only about 1% breaking down in 100 years, which harms the environment [*]. Therefore, I suggest modifying your statement in section 1.3.

*Rossi V., Cleeve-Edwards N., et al Life cycle assessment of end-of-life options for two biodegradable packaging materials: Sound application of the European waste hierarchy. J. Clean. Prod. 2015;86:132–145. doi: 10.1016/j.jclepro.2014.08.049. 

3. In Section 2.4, include a schematic illustrating the whole manufacturing process.

4. In line 201, please specify the temperatures for each of the four zones required in PLA-limestone powder filament production. This information can help other researchers reproduce it more easily.

5. In Section 2.5, include the relevant references for the EN ISO 527-2:2019 and EN ISO 179 standards.

6. In line 294, does the image show the fracture surface or the polished surface? If it is an SEM image of the fracture surface, can you analyse the type of fracture?

7. In Fig. 4, it appears that a demonstrator produced from the PLA-LP filament. However, there is no table provided with the printing parameters (e.g., nozzle temperature, bed temperature, printing speed, infill percentage, etc.).

8. It appears that the 25% and 50% LP weight fractions are too high to improve the mechanical properties, as they have worsened the mechanical strenght of the injection-moulded parts. It would have been better to start with lower fractions, such as 2% and 5% LP.

9. Although a printed demonstrator was produced, I do not see the mechanical results of the standard printed samples in your study.

Comments on the Quality of English Language

The English could be improved to more clearly express the research.

Round 2

Reviewer 2 Report

Comments and Suggestions for Authors

My first comment has not indicated in the paper. Using the high percentage of LP in the PLA which decline mechanical performance is not repurposing. Furthermore, the mechanical results of the printed samples have not reported. So, I recommend rejecting the paper.

Comments on the Quality of English Language

The English can be improved.

Author Response

Comments and Suggestions for Authors

My first comment has not indicated in the paper. Using the high percentage of LP in the PLA which decline mechanical performance is not repurposing. Furthermore, the mechanical results of the printed samples have not reported. So, I recommend rejecting the paper.

R: Dear Reviewer,

Thank you for your feedback. I would like to address your comments as follows:

Regarding your first comment, it appears to reiterate a point that I have already addressed in my initial response. Following the editor's guidance, I was instructed not to reconsider this point, as it had been clarified previously.

The purpose of this work was explicitly stated and supported by the tests we conducted, which demonstrate that the composite functions effectively as a "green" polymer. The necessity to compare with printed specimens in this particular article seems unwarranted, given that our results confirm the material behaves similarly to any standard polymer. While printing test specimens is indeed a priority for future work, as explained earlier, it is currently unfeasible due to practical constraints.

Comments on the Quality of English Language

The English can be improved.

R: As for the quality of the English language, I would appreciate it if you could specify which sections or phrases you found unclear or challenging to understand. Having lived in England for several years, my English may reflect regional variations or nuances, which could differ from what was anticipated. I am committed to ensuring the clarity and quality of the manuscript and would be happy to make improvements based on your detailed suggestions.

Thank you for your time and consideration.

Best regards,

Dora